# Structure and chemistry of graphene oxide in liquid water from first principles

Félix Mouhat[1], François-Xavier Coudert [2] & Marie-Laure Bocquet [1✉]

Graphene oxide is a rising star among 2D materials, yet its interaction with liquid water remains a fundamentally open question: experimental characterization at the atomic scale is difficult, and modeling by classical approaches cannot properly describe chemical reactivity. Here, we bridge the gap between simple computational models and complex experimental systems, by realistic first-principles molecular simulations of graphene oxide (GO) in liquid water. We construct chemically accurate GO models and study their behavior in water, showing that oxygen-bearing functional groups (hydroxyl and epoxides) are preferentially clustered on the graphene oxide layer. We demonstrated the specific properties of GO in water, an unusual combination of both hydrophilicity and fast water dynamics. Finally, we evidence that GO is chemically active in water, acquiring an average negative charge of the order of 10 mC m$^{-2}$. The ab initio modeling highlights the uniqueness of GO structures for applications as innovative membranes for desalination and water purification.

[1] PASTEUR, Département de Chimie, École Normale Supérieure, PSL University, Sorbonne Université, CNRS, 24 Rue Lhomond 75005 Paris, France. [2] Chimie ParisTech, PSL University, CNRS, Institut de Recherche de Chimie Paris, 75005 Paris, France. ✉email: marie-laure.bocquet@ens.fr

Graphene oxide (GO) is a graphene-based material that has gained significant interest in the last two decades[1,2] due to its straightforward, scalable, and low-cost synthesis. It has been proposed for numerous applications: for instance, GO is a promising material as an efficient sieve for water remediation[3–7] and for sustainable energy production via fuel cells[8,9]. Such properties strongly depend on the chemical groups present at the surface and versatile synthesis conditions using different reagents and oxidation durations have been proposed to allow a slight variation of the chemical composition of the material[10–13]. Consequently, there is a large variety of GO surfaces with a typical O/C ratio varying from 28 to 36%[14]. This ratio is known to decrease down to O/C = 5–10% when a thermal exfoliation is applied to the GO to produce single sheets of functionalized graphene, as some chemical functions are removed[15,16]. Using different characterization techniques, such as solid NMR, XPS, or Raman spectroscopy measurements, many theoretical models attempting to describe the GO surface have been proposed so far. The most widely used model is the one proposed by Lerf and Klinowski[17,18] originally for graphite oxide, and which has since then been largely used in the literature to describe monolayers of GO. Within this model, the layer is depicted as a random distribution of flat aromatic regions with unoxidized benzene rings, wrinkled regions containing hydroxyl or 1,2-ethers (epoxide) groups, and carboxylic acids grafted on the edges of the sheet. Despite some further refinements, such as the anti position of the hydroxyl pairs in the basal plane[19,20], there is to the best of our knowledge, no clear statement about the spatial arrangement of the oxidized functions along the graphene layer. Are they randomly distributed along the surface or not? What are the consequences on the stability of the material and its chemical properties, especially when it interacts with water? How does that impact filtration properties across GO membranes?

Indeed, the interaction of GO with water is particularly important due to the observed superpermeability of GO membranes to water molecules and its understanding requests modeling at the atomic scale. To date there are mostly classical MD simulations, hence treating the oxygen chemical groups as passive in water[21–25].

In this manuscript, we propose a realistic model of the GO basal plane surface, describing both water and the chemical groups at the electronic level. Within this approach, we generate manifold GO replicas at a given O/C ratio, with functions randomly distributed or not over the surface. Our as generated GO layers are first studied without solvent, and among the stable ones (see below), six of them are placed in liquid water at room temperature by means of long ab initio molecular dynamics (AIMD) simulations. Next, we perform a thorough statistical analysis by averaging over time and over replicas in order to quantitatively measures some of the physical and chemical properties of valuable interest for the community working on 2D materials and water/solid interfaces.

## Results

**Realistic graphene oxide models**. We constructed realistic GO models initially in anhydrous conditions, based on an orthorhombic periodic supercell structure of graphene comprising 72 carbon atoms. Starting from this pristine basal plane of graphene, we grafted 18 chemical functions to carbon atoms: 12 hydroxyl groups and 6 epoxide functions, which corresponds to a 2:1 ratio. This choice is consistent with suggested chemical formulas[26] and recent theoretical studies[27]. By construction with periodic systems, we discard the edge functionalization with carboxylic functions, which are experimentally known to be in smaller proportion, as compared with hydroxyls and epoxide ones.

Hydroxyl groups are systematically placed in pairs, respecting an anticonfiguration (with the two groups on opposite sides of the sheet). The total number of functional groups is chosen so that there are 24 $sp^3$ carbon atoms linked to an oxygen atom (out of 72), and the functionalization rate corresponds to O/C = 18/72 = 25%, a value in the low range of typical GO flakes[28,29]. We note that the choice of O/C ratio results from a compromise: taking values closer to the higher experimental limit would have decreased the number of possible independent replicas (for the semiordered case because the number of C atoms bonded to an oxygen atoms acquire the majority status); while a lower ratio would require more simulation replicas in order to measure statistically significant changes in behavior.

With this procedure, we generated ten GO structures, with the exact same chemical composition. Furthermore, for half of the structures, herein referred to as "random models", the positions of the hydroxyl and epoxide groups were chosen randomly among all C atoms of the graphene sheet. For the other half structures, the groups were kept concentrated in one half of the graphene sheet. We call these the "semiordered models"— although the exact positions of the functional groups are stochastic, these models contain some correlation. By looking at the systems generated in each case (see Fig. 1), the semiordered models present by construction a nanoscale patterning, where the GO shows aromatic regions, percolating to form graphene-like wires, as was suggested in some past works[30,31].

For the ten GO structures displayed in Fig. 1, we first checked their structural stability and energetics in vacuum by carrying out a series of structure optimizations (see details in the "Methods"). We report in Table 1 the relative total energies, including both electronic and atomic contributions, of the various models after geometry optimization (atomic positions and cell parameters). The lowest-energy configuration was chosen as reference point, with $E_{ref} = -266.270$ eV per C atom. All models were sorted from most energetically stable to least stable, and an average energy was calculated for each type (random and semiordered). We first note that there is significant dispersion of the computed energies, and that it is more important for the random structures than for the semiordered ones—possibly due to the constraints on the chemical groups. We also remark that the semiordered structures are on average 222 kJ mol$^{-1}$ or equivalently 32 meV per C atom more stable than the random ones, hinting that the former type could be the dominant form of GO synthesized experimentally. This energetic stability is indeed substantial and, interestingly, compares well with the 40 meV per C atom stabilization offered by the adsorption of a graphene layer on a strongly coupling metal-like Ru[32,33].

Two different hypotheses, which do not exclude one another, can be formulated to account for this result. First, adding a chemical function on a carbon atom induces some strain around the carbon site. In the random structures, a significant part of the graphene lattice is stressed by the epoxide and alcohol functions, whereas in the semiordered case, some regions remain unperturbed, and are therefore lower in energy. There is an energetic gain to concentrate the defects in localized regions, instead of applying an homogeneous deformation. Secondly, in semiordered structures, the closest proximity of the oxygen-bearing functional groups leads to a higher number of internal hydrogen bonds stabilizing the edifice. Such findings have been suggested previously by theoretical work using different methodologies, such as cluster models studied at the Density Functional Theory (DFT) level[34], or larger systems modeled with reactive force fields[35].

For the remaining part of this paper, we will focus on the three most stable structures obtained for each type of model (marked with [a] in Table 1). Their chemical structure is sketched in Fig. 1,

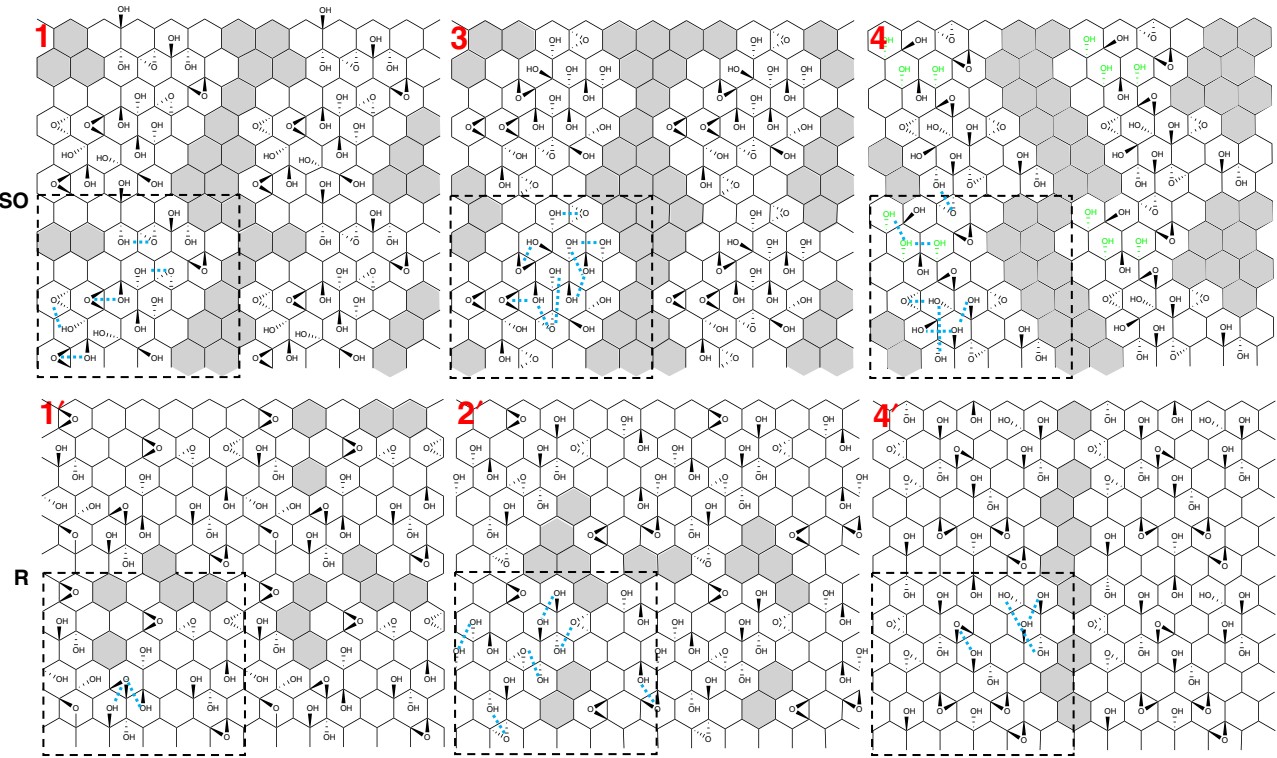

**Fig. 1 Models of graphene oxide studied in this work.** Semiordered (SO, top) and random (R, bottom) models of graphene oxide generated and studied as part of this work. Each material is designed according to a 25% O/C functionalization rate, and the oxygen-bearing groups are hydroxyls and epoxides with a 2:1 ratio. Gray hexagons are a guide for the eye, indicating the areas of pristine graphene, where no chemical function is grafted. Double C=C bonds are not drawn as double bonds, for the sake of clarity. All the configurations are represented as a 2 × 2 supercell of the graphene oxide model (the cell size is indicated in dotted lines on the top-left panel). Numbering of the models corresponds to Table 1. Intramolecular hydrogen bonds in GO are indicated by light-blue dotted lines, and the special reactive motif of three consecutive hydroxyl groups in the semiordered 4 model is colored in light green.

**Table 1 Relative total energy $E - E_{ref}$ (in meV per carbon atom) of the ten generated semiordered or random GO structures with respect to the most stable system, labeled "ref".**

| GO configuration | Electronic energy (meV per C atom) |
|---|---|
| Semiordered 3[a] (ref) | 0 |
| Semiordered 4[a] | 3 |
| Semiordered 1[a] | 13 |
| Semiordered 5 | 13 |
| Semiordered 2 | 33 |
| Average for semiordered | 16 |
| Random 2′[a] | 12 |
| Random 1′[a] | 27 |
| Random 4′[a] | 40 |
| Random 5′ | 72 |
| Random 3′ | 81 |
| Average for random | 48 |

All structures are relaxed using the CP2K code (see "Methods").
[a]The three most stable structures of each GO type that are further studied in this paper and displayed in Fig. 1.

where gray hexagons are used as guide for the eye to indicate the remaining regions of pristine graphene in GO models. It becomes clear that such regions are scarce and small in the random models (bottom panel), while in the semiordered models there are large pristine sectors that percolate to form one-dimensional wires.

**Structural properties**. In the following, we analyze the structural properties of the selected semiordered and random GO configurations, first in vacuum, and then in presence of liquid water. These properties were obtained through ab AIMD at room temperature (see Supplementary Fig. 2 and the "Methods" for details). The size of the simulation box along the $z$-axis, perpendicular to the GO sheet, imposes the scale of confinement of the water. We chose a value ($c \simeq 14.5$ Å) in good agreement with the typical interlayer spacing for GO laminates that swell in water (~14 Å)[36,37]. Extending first the stability analysis in the presence of water, the conclusions obtained in vacuum hold true: on average the stability of semiordered GO layers compared with random GO ones is increased, from 2.4 eV in vacuum to 3.3 eV in water (see details in Supplementary Table 1). Next, we looked at four characteristic distances and angles, displayed in Fig. 2. In a nutshell, the histograms of C–C distances and the $\widehat{CCC}$ angles indicate how the original graphene layer is perturbed by the presence of chemical functions (alcohols and epoxides). Moreover, the distribution of the $\widehat{COC}$ and the $\widehat{COH}$ angles allows us to look at the structure of the functional groups themselves, and to see how the GO interacts with the surrounding $H_2O$ molecules.

We first point out that the $d_{C-C}$, $\widehat{CCC}$ and $\widehat{COC}$ distributions for either the semiordered and random GO models do not significantly change with the environment, i.e., in vacuum or in the presence of water. Panels a–c of Fig. 2 are therefore only shown in water. As expected the C–C distance (Fig. 2a) is elongated by the presence of oxidizing groups with respect to the reference distance in graphene, $d_{C-C,graphene} = 1.42$ Å[38]. Moreover, the semiordered and random models of GO give markedly

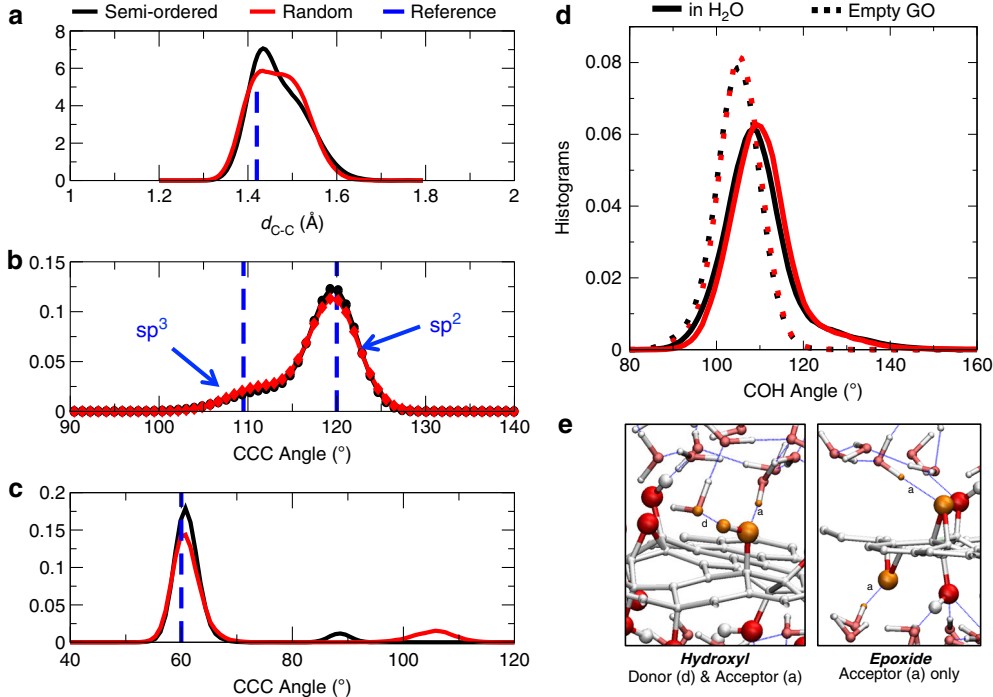

**Fig. 2 Structural properties of graphene oxide.** Histograms of the C–C distance $d_{C-C}$ (**a**), the $\widehat{CCC}$ angle (**b**), and the $\widehat{COC}$ angle (**c**) for semiordered (black) and random (red) graphene oxide models. Blue dashed lines correspond to the values in reference systems: $d_{C-C} = 1.42$ Å in graphene, $\widehat{CCC}_{sp^3} = 109.28°$, $\widehat{CCC}_{sp^2} = 120°$, and $\widehat{COC} = 60°$ for epoxide. **d** Histogram of the $\widehat{COH}$ angle for the GO in vacuum (dashed lines) and GO solvated in water (solid lines). **e** Snapshots of the different H bonds (visualized as blue dashed lines) types classified in Table 3. The atoms of each type of chemical function (hydroxyl group or epoxide) involved in the H bonds between the surface and $H_2O$ are highlighted in orange.

different distributions. There is a more pronounced peak around the graphene reference value for the semiordered configuration (black curve), corresponding to the regions that are not functionalized (i.e., pristine or graphene-like), whereas the distribution is more diffuse in the random case (red curve), as the chemical functions are spread over the entire GO sheet.

In contrast the $\widehat{CCC}$ angles distributions (Fig. 2b) coincide for the semiordered and random models, exhibiting two peaks corresponding to $sp^2$ and $sp^3$ carbon atoms, with different peak heights. We tried to fit these distributions by two Gaussians:

$$f(x) = \frac{\lambda}{\sqrt{2\pi\sigma_{sp^3}^2}} e^{-\frac{(x-\mu_{sp^3})^2}{2\sigma_{sp^3}^2}} + \frac{1-\lambda}{\sqrt{2\pi\sigma_{sp^2}^2}} e^{-\frac{(x-\mu_{sp^2})^2}{2\sigma_{sp^2}^2}}, \quad (1)$$

where symbols (black circles and red diamonds) in Fig. 2 correspond to the fit. The main fitting parameter $\lambda$ is an estimation of the functionalization rate O/C, which we expect to be around $\lambda_0 = 0.25$ (25% $sp^3$ C atoms) if the model is appropriate. We report in Table 2 the parameters obtained from this analysis. A very good agreement is found between $\lambda_{fit} = 0.22$–0.23 and $\lambda_0$. The values of $\mu_{sp^3}$ and $\mu_{sp^2}$ are in good agreement with the typical values tabulated for $sp^3$ and $sp^2$ angles, confirming that the $\widehat{CCC}$ angles distribution can be a useful tool to determine accurately the functionalization rate of GO. While such angle distributions may not be directly accessible experimentally, other quantities can be correlated to these angle distributions, such as bending vibration frequencies.

The distributions of epoxide $\widehat{COC}$ angles, displayed in Fig. 2c, are sharp and well-defined, yet at the same time slightly shifted to values > 60°. Despite their stability, these functions are thus likely to interact with neighboring –OH groups or $H_2O$ molecules. We also note the presence of small, broad peak at unexpectedly large

**Table 2 Fitting parameters of the $\widehat{CCC}$ angle distributions for the semiordered and the random graphene oxide models, using the two Gaussian functions of Eq. (1).**

| Fitting parameter | Semiordered | Random |
|---|---|---|
| $\lambda$ | 0.23 | 0.22 |
| $\mu_{sp^3}$ | 112.2 | 111.4 |
| $\sigma_{sp^3}$ | 4.3 | 3.6 |
| $\mu_{sp^2}$ | 119.7 | 119.6 |
| $\sigma_{sp^2}$ | 2.6 | 2.8 |

values the $\widehat{COC}$ angle. These correspond to specific situations, discussed in detail in the Supplementary Information, such as the formation in rare cases of a 1,3-epoxide (Supplementary Fig. 1) after rearrangement of the structure at $T = 0$ K (see Supplementary Fig. 2). Large $\widehat{COC}$ angles are also seen in cases of strongly strained structures where two epoxides are near-neighbors on a same hexagonal pattern, inducing large local stress that stretches the involved C–C distances and therefore flattens the corresponding epoxides (see Supplementary Fig. 3).

Finally the $\widehat{COH}$ angle distribution (Fig. 2d) is narrow and symmetric around $\widehat{COH} = 105°$ in vacuum, relatively unaffected by the order of the functions. In sharp contrast, in the presence of water the peak shifts to higher values and displays a broad tail—with values up to $\widehat{COH} = 160°$ being observed. This can be understood as a competition between intra- and intermolecular hydrogen bonds: in vacuum, to minimize its internal energy, some H bonds are formed between alcohols and epoxides functions of the GO. In the presence of water, in addition to the already present intramolecular H bonds, the GO can form new

intermolecular H bonds, between its oxygen-containing groups and the $H_2O$ molecules of the liquid. This competition is a first indicator of the presence numerous H bonds between the GO sheet and the liquid water, suggesting that the GO might be reactive.

**Hydrogen bonds and dynamics**. We therefore performed a systematic analysis of the hydrogen bonds present, differentiating between donor (D) and acceptor (A) atom types using Chemfiles/cfiles[39,40]. To do so, we adopted selection criteria typical for H bonds of moderate strength, as they mostly are in proteins[41]: from the trajectory, we consider a potential H bond as formed if the D–A distance $d_{DA} \leq 3.5$ Å, and $\widehat{DHA} \leq 30°$. Figure 2e represents a zoom-in view of an MD snapshot focusing on the H bonds (marked in dotted blue lines) involving hydroxyl groups (left) and epoxide functions (right) of GO and interfacial water molecules.

In our analysis, we assume that two H bonds per epoxide (two lone pairs) and three per alcohol (two lone pairs + one donor site) can be formed with surrounding $H_2O$ molecules. Therefore, the theoretical maximum number of water–GO hydrogen bonds in our systems is $2 \times 6 + 3 \times 12 = 48$. Table 3 shows that a large fraction of the functional groups of the GO (roughly 65%) are directly involved in a H bond, either with a neighboring function or with a $H_2O$ molecule. As expected, the fraction of intramolecular H bonds is more important in the semiordered models than in the random structures since the functional groups are closer to each other in the highly functionalized regions. Because of the numerous lone pairs on oxygen atoms, the GO sheet accepts more H bonds that it donates. However, regarding the occupancy rates of the donor/acceptor sites of the surface in tghe right side of Table 3, epoxides seem to be discarded from the counting and hence less cooperative than hydroxyl groups. Almost all the protons at the GO surface are involved in a H bonds, at variance with the oxygens sites which are partially participating. Indeed, the oxygen position is more constrained, leaving less flexibility for the spontaneous formation/breaking of H bonds with surrounding $H_2O$ molecules.

Finally, we turn to the impact of the GO on the transport properties of the liquid water. We estimated the water diffusion coefficients for semiordered GO models, and compared them with an ab initio MD simulation of bulk water in the same conditions (see Supplementary Fig. 9 and Supplementary Table 2). As displayed in Supplementary Fig. 9, we find that the average lateral diffusion value for water near the semiordered GO is around $0.67 \times 10^{-6}$ cm$^2$ s$^{-1}$ and a similar value for bulk water. This is an important and surprising conclusion, in stark contrast with the previous literature on confined water: contrary to other confined systems at 1–2 nm scale[42,43], the water dynamics near the GO sheets is not slowed down, but its transport is as fast as in the bulk liquid. This occurs despite strong hydrogen bonding

between the water and the GO, hinting at the very dynamic nature of such H bonds.

**Reactive processes at the graphene oxide/water interface**. As shown above, there are numerous hydrogen bonds between GO and water: they are natural sites at which chemical reactions may be initiated. We analyzed the ab initio MD trajectories of semiordered and random GO models by dynamically checking the coordination of each oxygen atom of the GO sheet. Hence, we could identify three types of reactive events that we detail now.

Figure 3 illustrates two such processes: the ring opening of an epoxide (a) and the deprotonation of a hydroxyl group (b). In the first case, the C–O bond breaking leads to a zwitterionic form of the opened epoxide, with a negatively charged alcoholate group accompanied by a positive charge on an adjacent carbon atom. This process does not create net charge on the GO, unlike the second process observed, where deprotonation yields a negative charge on the GO sheet, balanced by an excess proton in the liquid water (hydronium cation). These two types of chemical events are the common ones, occurring in all our MD simulations, regardless of the GO model. We highlight, however, that we did not observe disruption of the C–C bond of an epoxide leading to a 1,2-ether function, as proposed in a recent study[44].

We also observe a third type of chemical reaction, much rarer, depicted in Fig. 4. In one of the semiordered models, the distribution of OH groups results in a close proximity of three hydroxyls, with two intramolecular hydrogen bonds between them. This chain of H bonds is suitable for proton transfer as in the Grotthuss mechanism[45], and indeed, we observed the proton of one OH group jumping to the next, forming an adsorbed water molecule and an alcoholate. The desorption of this adsorbed water molecule into the liquid is a dehydration reaction and is concerted with the transformation of the alcoholate into an epoxide. This highlights the possible lability over time of the oxygen-bearing functional groups in hydrated GO, affecting both the C/O ratio and the distribution of hydroxyl and epoxide groups. The structural motif that results in dehydration (highlighted in light green in Fig. 1) is only present once in our models, in the semiordered model 4. In other models, three consecutive hydroxyls pointing to the same direction can sometimes be identified but, unlike in model 4, they are participating in other intramolecular H bonds with neighboring epoxide groups (see Fig. 1).

Moreover, in contrast with simulations in liquid water, reactive events are scarce for GO replicas simulated in vacuum: no surface hydroxyl deprotonation, less frequent epoxide opening. For the semiordered model 4 exhibiting the reactive motif, we observe that dehydration also occurs in vacuum, but at a longer time (see Supplementary Section II.1 and Supplementary Fig. 3 for details).

**Charge of graphene oxide in water**. We now aim at quantifying these chemical events over time, and measure the net charge carried by the GO in presence of water. Figure 5 displays the time evolution of the presence of neutral and charged species, both in the GO sheet and the interfacial water, for the three semiordered GO models (data for three random models are displayed in Supplementary Fig. 1). Neutral species are hydroxyl and epoxide groups, while charged species are alcoholate groups and hydronium cations. We first observed that, over the course of our MD simulations, the three different semiordered replicas behave very differently over time—this shows the impact of functional group distribution on the properties of the GO, and justifies the need for replicas in modeling GO. It is noticeable that all replicas start with five epoxides and not six showing that the initial geometry optimization process has resulted into one epoxide opening,

**Table 3 Average percentage of intra- and intermolecular hydrogen bonds between graphene oxide and the surrounding $H_2O$ molecules, and donor/acceptor sites occupancy analysis of the intermolecular H bonds.**

| | | Semiordered | Random |
|---|---|---|---|
| % H bonds | Intermolecular | 49 | 57 |
| | Intramolecular | 11 | 7 |
| | Total | 60 | 64 |
| % occupancy | Donor | 70 | 79 |
| | Acceptor | 36 | 37 |

Each type of H bond is illustrated in Fig. 2e.

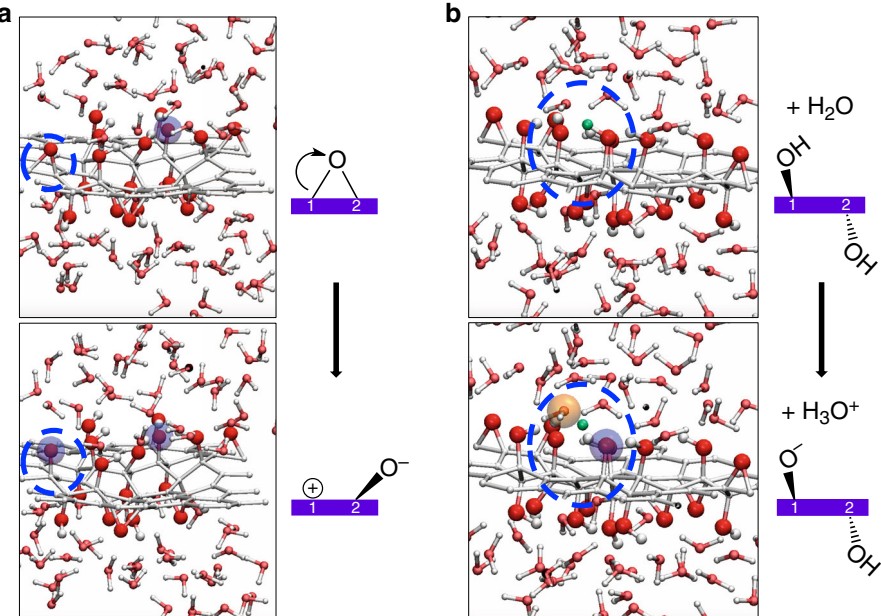

**Fig. 3 Snapshots taken from molecular dynamics simulations.** These snapshots are taken from trajectories of semiordered graphene oxide models in liquid water. **a** Opening of an epoxide function, leading to a zwitterionic form of the graphene oxide, but creating no net charge of the GO sheet. **b** Deprotonation of a surface hydroxyl group, leading to a surface alcoholate (blue shaded circles) and an excess proton (orange) in the liquid water. The labile hydrogen atom is highlighted in green. Schematics are presented on the right side, for clarity.

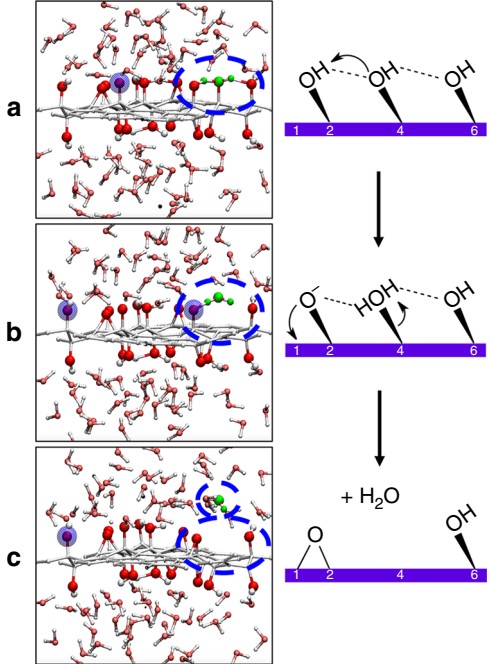

**Fig. 4 Spontaneous dehydration of the graphene oxide.** We highlight the proton transfer induced by the presence of a strong hydrogen bond network. **a**, **b**, and **c** are snapshots along the MD trajectory of the semiordered 4 model displayed in blue in Fig. 5. The leaving water molecule is colored in green. Schematics are presented on the right side, for clarity. See Fig. 3 for color scheme. This reactive motif of three consecutive hydroxyl groups linked by two intramolecular hydrogen bonds is only present in semiordered model 4 and colored in light green in Fig. 1.

reducing the strain in the material. The semiordered model 1 (black) remains moderately reactive with a few closing and opening of epoxide functions. The semiordered model 4 (blue curve) appears to be the most reactive one, and displays the dehydration mechanism described above. Finally the semiordered model 3 (red curve) is also reactive, displaying a few proton transfers and producing transient charged species. This latter model is the most energetically stable model (see Table 1).

We now turn to the impact of these chemical events on the charge of the GO layer. To do so, looking at the presence of alcoholate groups is not sufficient, as they can arise from both deprotonation and epoxide opening (which does not induce a net charge on the GO). Looking at the number of hydronium cations, which is by design the opposite of the net GO charge, the results are shown in Table 4. The average charge per replica for the semiordered models is negative, and its absolute value spreads from 2.5 to 13 mC m$^{-2}$—with an average value of 6 mC m$^{-2}$. These values, obtained from our ab initio simulations, compare favorably with recent measurements stating 16 mC m$^{-2}$[46] or with values extracted from zeta potential measurements at neutral pH being 11 mC m$^{-2}$ for GO and 3 mC m$^{-2}$ for reduced GO (rGO)[47]. It should be stressed that our GO replicas resemble more their rGO samples, as they contain no carboxylic functions. Moreover, random models yield different negative charges but remaining in the same order of magnitude (see Table S1).

It is an important feature that a GO sheet should no longer be modeled neutral when dipped into water, but instead carrying a modest negative charge, ranging from a few to tens of mC m$^{-2}$ and due to by partial deprotonation of its hydroxyl groups. It confirms that the p$K_a$ of GO is lower than the value of alcohols (16–19), and presumably lower than 14. This surface charge may explain electrostatic repulsion between flakes that induces their perfect layering at confined distances.

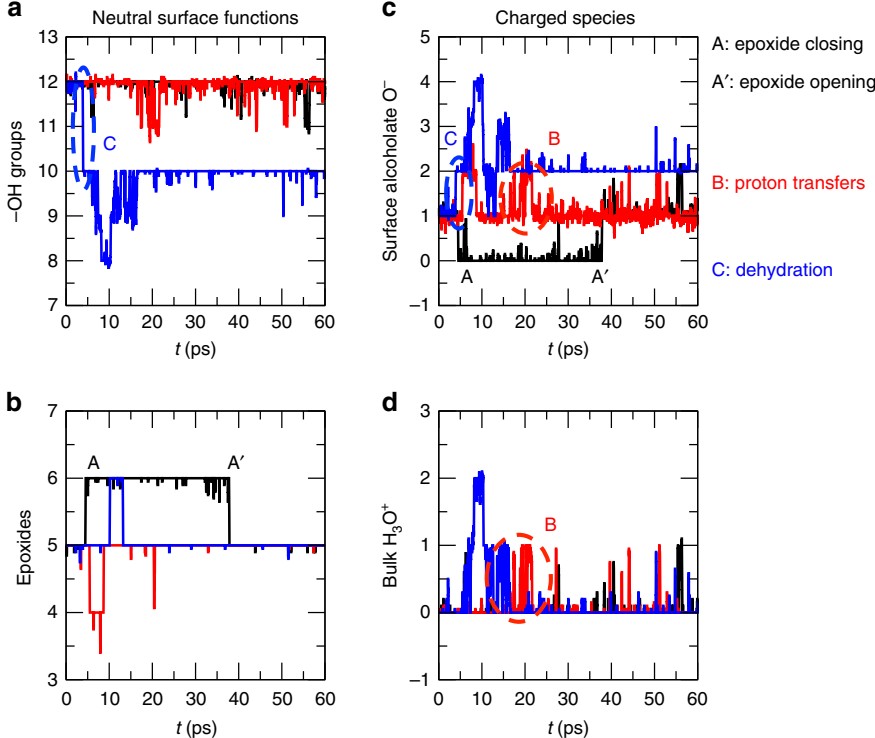

**Fig. 5 Dynamics of graphene oxide in liquid water.** Time evolution of the number of neutral functional groups of the GO surface (left) and charged species present at the GO surface or in the water bulk (right). **a** –OH groups. **b** Epoxides. **c** Surface alcoholate. **d** $H_3O^+$. The black, red, and blue curves are associated to the semiordered 1, 3, and 4 GO models, respectively. Some reactive events occurring in the GO dynamics in water are also identified and marked with capital letters.

**Table 4 Excess charge/GO surface (in mC m$^{-2}$) on the GO interfacing bulk water (equivalent with opposite sign to the excess charge distributed over the H$_2$O molecules of the bulk).**

| GO configuration | Excess surface charge (mC m$^{-2}$) |
|---|---|
| Semiordered 1 | −2.46 |
| Semiordered 3 | −4.27 |
| Semiordered 4 | −12.96 |
| Average | −6.56 |

## Discussion

In summary, by means of extensive ab AIMD simulations, we have built several realistic models of GO at a fixed O/C ratio, comprising hydroxyls and epoxide groups, and analyzed their respective stability, as well as physical and chemical behavior in liquid water. Our computational study emphasizes the importance of partly ordered models of GO in order to tackle its physical and chemical properties. Although computationally expensive, it is the combination of such models with the use of extensive ab initio MD that allows us to describe the chemical processes occurring at the GO/liquid water interface. In particular, we demonstrate the impact of functional group distribution on the GO sheet, and show that semiordered models—with correlated functional groups and some regions of pristine graphene—are the most stable structures in vacuum as well in liquid water. We also demonstrate the formation of a net negative charge on the GO induced by the reactivity with water, with a moderate charge in the same range of silica materials in water, but two orders of magnitude less than hexagonal boron nitride layers[48,49]. Altogether this combination may help understand the quite unique properties of GO in terms of water and ion transport, which are at the root of its use in water filtration and remediation. Our results highlight fast water dynamics, impeded neither by the confinement nor by the presence of hydrophilic oxygen groups. Notably, the favorable clustering of oxygen functions may open fast dynamic pathways for water transport on the remaining pristine graphene regions. This could explain the large hydrodynamic slippage of water across GO, as reported experimentally[37].

Because GO is experimentally very diverse in its chemical composition, and in the interlayer spacing of hydrated GO, a lot of perspectives emerge in ab initio modeling of GO in different conditions. In particular, the impact of the confinement thickness on the water dynamics is of interest for the future. The present work also opens the way for further studies on the impact of oxygen concentration (number of functional groups) on the GO/water interactions, by varying the O/C ratio and introducing carboxyl groups. The dynamics of charged ions at this charged interface, including hydronium and hydroxide ions, is also a wide open question, that quantum modeling at the electronic scale should be able to tackle.

## Methods

**Density functional theory (DFT) calculations**. All the simulations discussed in the paper and in this section have been performed at the ab initio level, using the DFT approach. DFT has therefore been used to both optimize the materials' geometry at zero temperature and to evaluate the ionic forces acting on each atom along the AIMD trajectories. Most of the calculations have been carried out with the CP2K software[50] and some preliminary electronic structure relaxations have been performed with VASP[51]. With C2PK, the DFT method is straightforwardly employed using the Quickstep module[52,53] of the package. As in ref. [54], DZVP-MOLOPT-SR-GTH basis sets were used[55] along with planewaves expanded to a 800 Ry absolute energy cutoff and a 40 Ry relative cutoff. Electronic cores were represented by Geodecker–Teter–Hutter pseudopotentials[56–58]. The Perdew, Burke, and Ernzerhof functional was used[59] with the DFT-D3 dispersion

correction scheme[60,61]. Technical details about the convergence of the calculation settings are provided in the Supplementary Information Section III.1.

**Construction of graphene oxide models**. In detail, we first built each empty semiordered or random GO replica by adding chemical functions according to the constraints detailed in the first part of the paper. Then, these structures have been placed into a $15 \times 13 \times 16$ Å$^3$ orthorhombic cell, corresponding to a $6 \times 6$ graphene-like lattice with an interlayer spacing of 16 Å. Then, we performed a preliminary geometry optimization and a cell relaxation using VASP. Indeed, adding chemical functions on the GO surface induces a mechanical strain and the lattice parameters $a$ and $b$ need to be adjusted. This procedure is then repeated with the C2PK code to ensure that the minimum energy configuration has been reach and to check the consistency of the results with two different codes. Once the empty surface is fully relaxed (i.e., the pressure of the system is about 10–100 MPa, a standard value for such materials), we generated a single 30 ps AIMD trajectory for each replica at $T = 300$ K, using periodic boundary conditions. The deuterium mass is substituted for all protons to reduce the timestep size needed for energy conservation in our Born Oppenheimer dynamics AIMD and to limit nuclear quantum effects of the proton. All the AIMD simulations were carried out with a 0.5 fs timestep in the $NVT$ ensemble, using a stochastic velocity rescaling (SVR) thermostat[62] with a time constant of 1 ps.

**Construction of hydrated graphene oxide models**. After having ensured the stability and studied the structural properties of the anhydrous GO structures, we embedded them into explicit water. This is achieved by adding 80 H$_2$O molecules using the PACKMOL program[63]. We imposed the water molecules to be distant of at least 2 Å from the GO surface to avoid steric overload due to the presence of hydroxyl/epoxide functions on the surface. The procedure is then very similar to the anhydrous case. First, a geometry and a simulation cell relaxation is necessary to make the H$_2$O molecules organized in a relevant chemical configuration. Let us notice that only the $c$ cell parameter is allowed to change along the $z$-axis to preserve the GO structure in the $xy$ plane. Later, a short AIMD trajectory of 1 ps at $T = 300$ K is with a small time constant of 100 fs for the SVR thermostat is generated to make the water liquid around the GO surface. Afterwards, a new cell relaxation is applied and a $\Delta c = 1.5$–1.8 Å variation is typically observed during this step. Finally, AIMD trajectories are generated after a 5 ps equilibration period: 60 ps long for each semiordered replica while we stopped the calculation at 30 ps for the random structures. A 60-ps-long trajectory of bulk liquid water at the experimental density $\rho = 0.99657$ g cm$^{-3}$ is also generated to discuss the diffusion coefficients on similar trajectories (see Supplementary Information Section II.9 for further details).

Finally, the observables presented in the paper are obtained using block averages of the AIMD trajectories of each material, using Chemfiles/cfiles[39,40].

## Data availability

Data supporting this study is available online on the repository at https://github.com/fxcoudert/citable-data. They can also be made available upon request to the corresponding author.

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

## Acknowledgements

F.M. and M.-L.B. acknowledge funding from the H2020-424 FETOPEN project NANOPHLOW number 766972. This work was granted access to the HPC resources of TGCC and CINES under grants A5-A0070807364 and A0070807069 by GENCI. We thank Vasu Kalangi and Lydéric Bocquet for insightful discussions about GO membranes, and Guillaume Fraux for his help with Chemfiles/cfiles.

## Author contributions

M.-L.B. designed the research. F.M. performed the simulations. F.M., F.-X.C., and M.-L.B. analyzed the data, wrote and revised the paper.

## Competing interests

The authors declare no competing interests.
