## [Peer Review File · Nature Communications]

Reviewers' comments:

Reviewer #1 (Remarks to the Author):

This work reports very interesting results concerning the structure of graphene oxide surfaces and their interactions with liquid water. The manuscript is very well written, and the results will be of interest for a large audience since such materials are investigated for many applications. I recommend publication in Nature Communications but I think the following points should mandatorily be addressed by the authors:

1 (minor): I think that the term "for the first time" should be avoided in the abstract. In addition I do not really see which gap is bridged in the same sentence.

2 (major): About the construction of the graphene oxide model:

-References 17 and 18 seem to be about graphite oxide. Don't the authors expect differences with graphene oxide?

-About the choice of having a functionalization rate of 25%: in the introduction the authors mention that it is of 5-10% when a thermal exfoliation is applied. Isn't that case the most relevant one?

- Page 9, when comparing their results with ref. 45, the authors mention the potential presence of carboxylic functions. Why didn't they consider them in their model?

3 (major): About the validation of the graphene oxide model: unless I missed something, the only argument for preferring semi-ordered structures over the random ones is based on an energy criterion. However I see in Table 1 that the difference is not so large, and in fact random 2' has an energy lower than 3 of the five semi-ordered structures. I can imagine that there are infinities of structure that can be generated with the same level of defects (and this would increase exponentially with the size of the sample, which here remains quite small) – even if it is large for DFT calculations!), how can the authors be sure that their data is enough to reach this conclusion? Wouldn't there be some experimental data (Raman, infra-red) that would allow them to have a more quantitative proof?

4 (major) Page 8, last paragraph: The authors conclude that the different replicas behave very differently. I think that this conclusion should be mitigated in view of the very short simulation times. Are they sure that similar reactions would occur if they used different starting points for each of the simulations?

Reviewer #2 (Remarks to the Author):

Bocquet and co-workers presented a computational study, using ab initio calculations, to study the structure and chemistry of graphene oxide (GO) in liquid water. This work studied GO in detail, and found that (i) the clustered (de-noted semi-ordered) GOs are more stable, relative to the random ones and (2) the similar/dissimilar structure behaviors including hydrogen bonding environments for studied GO structures in water, (iii) the fast diffusion of water on the GO surface, and (iv) the surface reactivity and its formed negative net charge. I think this work is well written and is evidently conducted with care. While there are still a few points that I think the authors should consider and address, my major concern is regarding the breakthrough achieved in this work to warrant a publication in Nat. Commun. Although this work helps bridge the gap in the understanding of GO materials, I do not see a breakthrough from this work to facilitate the development of GO in applications. I am therefore afraid that I do not find this work to be suitable to publish in Nat. Commun., but in a good physical journal such as JPC B or C. My recommendation is to reject this submission and suggest the authors to submit to other journals as suggested above.

A few other comments:

1. It would be important to also study the effect of oxygen concentration (number of functional groups and their ratios).

2. The authors commented on the stability based on the vacuum condition. Would the conclusion change if it is under solvated environment? Grossman and co-workers reported many years ago some evidences of "clustering of functional groups on GO/rGO materials (after annealing)". Maybe the results are in line with their observations?
3. It remains unclear how the surface reactivity relates to the detailed structure properties of GO. More discussion would make this work stronger.
4. What's the water diffusivity on the surface of random GO structures? Also, why the semi-ordered 4 has a noticeably smaller diffusivity than the semi-ordered 1 structure? Are water molecules mainly diffusing through the non-functionalized (sp²) part?

Reviewer #3 (Remarks to the Author):

The manuscript by F. Mouhat et al. reports the study of GO layers in water by means of ab initio molecular dynamics. The Authors find that GO sheets showing pristine graphene areas next to clusters of functional groups are the most stable configurations. Additionally, the GO layer is said to acquire a negative charge when immersed in water as a consequence of the release of protons.

The manuscript provides interesting insights into GO atomistic structure and reactivity. However, some of the conclusions (the clustering effect) were already proven to be independent on the surrounding media in literature while others (some reaction paths) might be also drawn for a sheet in absence of water. For this reason, the difference between GO in vacuum and in water should be highlighted with more explicit comparisons. The manuscript provides fertile ground for new very appealing analyses which might be implemented to enhance the impact of the paper. At the current state I would not recommend the manuscript for Nature Communications, but I would suggest major revisions.

Here are some comments on how to improve the quality of the manuscript:

1. One of the main points of the manuscript, the clusterization of the functional groups, was already proven both experimentally and theoretically [see the experimental study from Kumar et al. Nat. Chem. 6, 151 (2014) and the theoretical works: Carbon 100, 90 (2016), Sci. Rep. 3, 2484 (2013)] The main reason behind this behaviour is attributed, as indicated also by the Authors, to the strain induced by functional groups that is released if the functional groups move together in a suitable conformation [see, for instance, PRL 96, 176101 (2006)]. As such, I would rather focus the study on the effect that the presence of water has in the clustering process and I would perform more analyses about this.

2. In relation to the previous point, the Authors suggest a very interesting hypothesis on another possible drive for the clusterization: the maximization of the number of hydrogen bonds. This hypothesis deserves a more detailed analysis as it can be the key to understand the effect of water on GO. I would expect that the presence of water hinders the tendency for the functional groups to cluster as, when the sheet is submerged in water, hydrogen bonds form between the sheet and the H₂O molecules, rather than only among the GO functional groups.

A more detailed investigation on the water effect would quantify the consequences of intermolecular H bonding as opposite to intramolecular as, here, might lie the core for a different clustering process.

3. Authors should provide a direct comparison with their trajectories in vacuum as many of the chemical reactions occurring do not involve water molecules directly. For instance, the reactions described in Figure 3a and 4 might as well take place in vacuum.

4. I have some concerns about the methodology for the random generation of GO sheets. It was proven that epoxides and hydroxyls cluster in specific arrangements that lower the sheet

energy [see Sci. Rep. 3, 2484 (2013) and J. Phys. Chem. Lett. 10, 7492 (2019)]. Could the random generation lead to highly unstable sheets that might enhance the likeliness for functional groups to desorb?

For example, the reaction in Figure 4 that leads to a reduction of the sheet might be caused by the excessive strain on the GO sheet induced by the nearby functional groups.

What's Authors' opinion in this regard?

5. The different mechanisms shown in the paper are very interesting and deserve a deeper study, first of all, to identify whether there is an effect of the surrounding functional groups and secondly to understand what are the conditions that lead to such transitions.

Some non-dynamical studies about energy and barriers could reveal more information.

6. The negative charge of the GO sheet in water might be highly dependent on water pH at the beginning of the simulation.

The H⁺ electrochemical potential in water is low when the simulation starts (there is no H⁺ in solution) consequently there might be a high drive for the dissolution of the most movable species in GO following the prototypical reaction:

Which will continue until the chemical potential of the adsorbed H and the sum of electrochemical potentials of e⁻ and H⁺ are equal. This leads to an electrification of the GO sheet that is highly dependent on the initial density of H⁺ in solution.

How do Authors comment about this dependency?

POWER PDF TRIAL
www.kotax.com

Point-by-point response to the reviewers

Reviewer #1

This work reports very interesting results concerning the structure of graphene oxide surfaces and their interactions with liquid water. The manuscript is very well written, and the results will be of interest for a large audience since such materials are investigated for many applications. I recommend publication in Nature Communications but I think the following points should mandatorily be addressed by the authors:

1 (minor): I think that the term “for the first time” should be avoided in the abstract. In addition I do not really see which gap is bridged in the same sentence.

We have removed “for the first time”, and expanded the sentence to make it clearer in the abstract.

2 (major): About the construction of the graphene oxide model: References 17 and 18 seem to be about graphite oxide. Don't the authors expect differences with graphene oxide?

Indeed, references 17 and 18 discuss a graphite oxide model, based on experimental data from graphite oxide. However, they focus almost exclusively on the graphene oxide layer itself, and have been widely used in the literature as a reference model for both graphite oxide and graphene oxide. There are definitely structural and chemical differences between graphene oxide and graphene, as well as there are between graphene oxide with different chemical compositions and syntheses. The chemical composition we have taken in the model we developed in this work are, on the other hand, rather typical of graphene oxide.

We have added a note, after refs. 17 and 18 in the introduction part, to indicate that the Lerf and Klinowski models were originally developed for graphite oxide, and which has since then been largely used in the literature to describe monolayers of graphene oxide.

- About the choice of having a functionalization rate of 25%: in the introduction the authors mention that it is of 5-10% when a thermal exfoliation is applied. Isn't that case the most relevant one?

Indeed, the 25% of O/C ratio lies in the lower range of observed functionalization of GO samples at room T — but in the higher range compared to ratios observed after annealing. It should be pointed out that converting our O/C ratio into the number of C atoms affected by the O functionalization yields higher percentage, i.e. $(12 + 2 \times 6) = 24 / 72 = 0.33$ for the random case and $24/36 = 0.66$ (locally) for the semi-ordered case. Therefore, taking a model with a O/C ratio close to the higher limit (~ 0.375 corresponding to 18 OH and 9 epoxide that affect 27 C atoms) would have restricted the number of possible independent replicas for the very dense semi-ordered case (coverage of 0.75%).

On the other hand, taking a lower O/C ratio (closer to 5 %) would have made the impact of GO on the water smaller, requiring more replicas in order to measure statistically significant variations: both models (semi-ordered and random) would be closer in behavior.

We have added a comment in the model section : “We note that the choice of O/C ratio results from a compromise: taking values closer to the higher experimental limit would have decreased the number of possible independent replicas (for the semi-ordered case because the number of C atoms bonded to an oxygen atoms acquire the majority status); while a lower ratio would require more simulation replicas in order to measure statistically significant changes in behavior.”

- Page 9, when comparing their results with ref. 45, the authors mention the potential presence of carboxylic functions. Why didn't they consider them in their model?

This was done for the sake of simplicity: carboxylic groups are known experimentally to be in smaller concentration than –OH and epoxides. Moreover, carboxylic functions are present only at the edges of the graphene oxide sheet, and we have chosen not to create edges in our periodic models — introducing explicit edges in the model would make its size much larger than it currently is, and difficult to study with sufficiently long MD trajectories. Or, done on a smaller model, it would create an unrealistic edge/surface ratio, misrepresenting the actual chemistry of the graphene oxide.

We have added a comment on the draft in the model section: “By construction with periodic systems, we discard the edge functionalization with carboxylic functions, which are experimentally known to be in smaller proportion, as compared to hydroxyls and epoxide ones.”

3 (major): About the validation of the graphene oxide model: unless I missed something, the only argument for preferring semi-ordered structures over the random ones is based on an energy criterion. However I see in Table 1 that the difference is not so large, and in fact random² has an energy lower than 3 of the five semi-ordered structures. I can imagine that there are infinities of structure that can be generated with the same level of defects (and this would increase exponentially with the size of the sample, which here remains quite small – even if it is large for DFT calculations!), how can the authors be sure that their data is enough to reach this conclusion? Wouldn't there be some experimental data (Raman, infra-red) that would allow them to have a more quantitative proof?

We agree with the referee that the data obtained within this work is not sufficient, in and of itself, to reach the conclusion that energetic stability is systematically favorable for semi-ordered structures. We note that this direct observation of the energetics on realistic models by first-principle methods is consistent with the results obtained theoretically and computationally by other methods, that we quote in our revised manuscript. On this, see also the discussion to reviewer #3, point 1).

This comment, as well as the remarks from reviewer #3 (point 2) have also prompted us to look into the impact of hydration on the energetic stability of our GO models: the extra analysis performed was added to the supplementary information (Table S2) and is discussed below.

4 (major) Page 8, last paragraph: The authors conclude that the different replicas behave very differently. I think that this conclusion should be mitigated in view of the very short simulation times. Are they sure that similar reactions would occur if they used different starting points for each of the simulations?

This is a very interesting remark, and it has prompted us to consider the impact of the initial configuration of the water molecules on the properties of the hydrated graphene oxide models. In order to probe this effect directly, we have performed additional *ab initio* MD simulations on one of our models (semi-ordered model 3). For this GO model, we have applied our methodology a second time, generating a different random initial packing of the same number of water molecules. We have added these results in Supplemental Information (Figure S8 and related discussion). The main conclusion is that, although the details of the dynamics are different (because the initial configurations are different), the overall behavior is rather close: opening of one epoxide and associated formation of one surface alcoholate on average.

Reviewer #2

Bocquet and co-workers presented a computational study, using *ab initio* calculations, to study the structure

and chemistry of graphene oxide (GO) in liquid water. This work studied GO in detail, and found that (i) the clustered (de-noted semi-ordered) GOs are more stable, relative to the random ones and (2) the similar/dissimilar structure behaviors including hydrogen bonding environments for studied GO structures in water, (iii) the fast diffusion of water on the GO surface, and (iv) the surface reactivity and its formed negative net charge. I think this work is well written and is evidently conducted with care. While there are still a few points that I think the authors should consider and address, my major concern is regarding the breakthrough achieved in this work to warrant a publication in *Nat. Commun.* Although this work helps bridge the gap in the understanding of GO materials, I do not see a breakthrough from this work to facilitate the development of GO in applications. I am therefore afraid that I do not find this work to be suitable to publish in *Nat. Commun.*, but in a good physical journal such as *JPC B* or *C*. My recommendation is to reject this submission and suggest the authors to submit to other journals as suggested above.

A few other comments:

1. It would be important to also study the effect of oxygen concentration (number of functional groups and their ratios).

The computational methodology we have developed in this work, and presented in the manuscript, can be applied to systems with slightly different characteristics: different O/C ratio, different ratio of epoxides/OH groups, presence of carboxylic groups (see discussion of this in response to reviewer #1's comment). However, these first-principles MD simulations are computationally costly, because they describe the system at the quantum chemical level (with DFT) and sample the dynamics of the systems studied. We therefore needed to make a choice in the systems under study, and have preferred in this initial study to focus our HPC time to obtain multiple replicas of each type of model (to make sure our results were robust), rather than test different chemical compositions.

We have added some discussion in the conclusion about the perspectives opened by this work, and in particular, about the interest in studying the influence of O/C ratio and in particular the low O/C ratio around 5%.

2. The authors commented on the stability based on the vacuum condition. Would the conclusion change if it is under solvated environment? Grossman and co-workers reported many years ago some evidences of "clustering of functional groups on GO/rGO materials (after annealing)". Maybe the results are in line with their observations?

We agree with the referee that the results obtained in this work on energetic stability, done in vacuum, do not preclude the possibility of an effect of the solvation environment. What we have seen in our simulations is that the presence of solvent, and the relaxation of the GO sheet that it induces, does not change the overall conclusions regarding stability. In this, our results agree with the findings of Grossman (DOI: 10.1016/j.carbon.2015.12.087).

We have added this reference, and related discussion, to the main text of the manuscript (page 4). We have also added a comparison of energies under solvation in the Supplementary Information (Table S2).

3. It remains unclear how the surface reactivity relates to the detailed structure properties of GO. More discussion would make this work stronger.

We thank the reviewer for this suggestions, which is also found in some comments of the two other reviewers. We have identified a specific structural motif with reactive properties, namely the presence of three consecutive -OH groups, pointing toward the same direction, and linked by hydrogen bonds. This motif gives rise to dehydration reaction of the middle -OH group, seen both in vacuum and facilitated in the hydrated GO.

4. What's the water diffusivity on the surface of random GO structures? Also, why the semi-ordered 4 has a noticeably smaller diffusivity than the semi-ordered 1 structure? Are water molecules mainly diffusing through the non-functionalized (sp^2) part?

We have tried to find a simple chemical explanation for the difference in water diffusion between the different semi-ordered models. We have looked at the analysis of the structures themselves and the presence of intramolecular H bonds (Table S1), as well as the number of non-functionalized graphene rings and diameters of the non-functionalized regions. We have not been able to establish any direct correlation between those parameters and the diffusion coefficients. Moreover, since diffusion is a collective phenomenon, we cannot separate the contributions from the water molecules near the non-functionalized sp^2 regions.

Moreover, given the relative short simulations times imposed by the high computational cost of ab initio molecular dynamics, we do not want to overinterpret the differences between the semi-ordered models, but focus on the average picture, which is the relatively fast dynamics of the water near the GO sheets, in comparison with the slowdown expected at this level of confinement.

Reviewer #3

The manuscript by F. Mouhat et al. reports the study of GO layers in water by means of ab initio molecular dynamics. The Authors find that GO sheets showing pristine graphene areas next to clusters of functional groups are the most stable configurations. Additionally, the GO layer is said to acquire a negative charge when immersed in water as a consequence of the release of protons. The manuscript provides interesting insights into GO atomistic structure and reactivity. However, some of the conclusions (the clustering effect) were already proven to be independent on the surrounding media in literature while others (some reaction paths) might be also drawn for a sheet in absence of water. For this reason, the difference between GO in vacuum and in water should be highlighted with more explicit comparisons. The manuscript provides fertile ground for new very appealing analyses which might be implemented to enhance the impact of the paper. At the current state I would not recommend the manuscript for Nature Communications, but I would suggest major revisions.

Here are some comments on how to improve the quality of the manuscript:

1. One of the main points of the manuscript, the clusterization of the functional groups, was already proven both experimentally and theoretically [see the experimental study from Kumar et al. Nat. Chem. 6, 151 (2014) and the theoretical works: Carbon 100, 90 (2016), Sci. Rep. 3, 2484 (2013)] The main reason behind this behaviour is attributed, as indicated also by the Authors, to the strain induced by functional groups that is released if the functional groups move together in a suitable conformation [see, for instance, PRL 96, 176101 (2006)]. As such, I would rather focus the study on the effect that the presence of water has in the clustering process and I would perform more analyses about this.

We agree with the reviewer that the existence of clustering of oxygen groups is not new, but would note that from the computational point of view, they have been previously studied in most cases with simpler cluster models at the DFT level (Zhou & Bongiorno, *Sci. Rep.* 2013), or with less accurate reactive force fields (Kumar et al, *Carbon* 2016). Moreover, although we are satisfied to find this behavior reproduced, we mostly use it as a starting point for further analyses of the properties of the hydrated GO (which were further extended as part of the review process).

We have added a sentence to the discussion on page 4 about the clustering of oxygen group, to highlight the previous works on this topic, as suggested by the reviewer.

2. In relation to the previous point, the Authors suggest a very interesting hypothesis on another possible drive for the clusterization: the maximization of the number of hydrogen bonds. This hypothesis deserves a more detailed analysis as it can be the key to understand the effect of water on GO. I would expect that the presence of water hinders the tendency for the functional groups to cluster as, when the sheet is submerged in water, hydrogen bonds form between the sheet and the H₂O molecules, rather than only among the GO functional groups.

A more detailed investigation on the water effect would quantify the consequences of intermolecular H bonding as opposite to intramolecular as, here, might lie the core for a different clustering process.

We thank the reviewer for this suggestion, which is very interesting. Indeed, we had not initially looked at the impact of hydration on the energetics of the GO models. We have performed further calculations, and have inserted a new Table S2 to the Supplementary Information. This analysis shows that the relative energetic stability of random vs. semi-ordered models is not changed upon hydration. We find that the energy difference (on average) between semi-ordered and random models is changed from 2.45 eV (without water) to 3.27 eV (hydrated).

We have added a few sentences about this analysis in the “Structural properties” section, on page 5.

3. Authors should provide a direct comparison with their trajectories in vacuum as many of the chemical reactions occurring do not involve water molecules directly. For instance, the reactions described in Figure 3a and 4 might as well take place in vacuum.

This is a very valid point, and in order to probe the behavior of our GO models in vacuum, we have run them further for 60 ps of production dynamics. These longer simulations have revealed a very interesting behavior: the reactive motif identified in semi-ordered model 4 can also lead to dehydration (loss of a water molecule) in the GO in vacuum, but that event only occurs after 40 ps of dynamics, while the reaction is fast in the hydrated model. We discuss this comparison of hydrated GO and empty GO in the revised manuscript, on page 9, and in the supplementary information (Figure S3 and related discussion, page S7).

4. I have some concerns about the methodology for the random generation of GO sheets. It was proven that epoxides and hydroxyls cluster in specific arrangements that lower the sheet energy [see Sci. Rep. 3, 2484 (2013) and J. Phys. Chem. Lett. 10, 7492 (2019)]. Could the random generation lead to highly unstable sheets that might enhance the likeliness for functional groups to desorb? For example, the reaction in Figure 4 that leads to a reduction of the sheet might be caused by the excessive strain on the GO sheet induced by the nearby functional groups. What's Authors' opinion in this regard?

The randomness in the generation of the models is, we think, a key component in order to make sure that the models are as realistic as possible, i.e. that we do not introduce specific features artificially. We agree with the reviewer that, because of this random element in the generation of models, it could lead in some cases to highly unstable GO sheets. We have, however, taken steps to avoid this: first, only models that are stable upon initial energy minimization have been considered (i.e., the models are not spontaneously reactive). Secondly, we have performed MD simulations only on some, and not all, of the models generated: we selected these models by energetic criteria, therefore eliminating potentially highly unstable model. Finally, we have run simulations on several models of each type (replicas) in order to study the diversity in behavior, and to obtain physically meaningful averages that do not depend only on a single simulation.

5. The different mechanisms shown in the paper are very interesting and deserve a deeper study, first of all, to identify whether there is an effect of the surrounding functional groups and secondly to understand what are the conditions that lead to such transitions. Some non-dynamical studies about energy and barriers could reveal more information.

Using both the simulations of the hydrated models, and the new simulations of the models in vacuum, we have been able to identify a specific reactive motif in the GO structure, that leads to the dehydration reaction. We have marked it in Figure 1, and discuss it in the main text on page 9, and in the supplementary information (page S7).

As suggested by the reviewer, we have used the MD simulations to estimate the energetics of the dehydration reaction. The results were added to the supplementary information (page S9 and Figure S4). We find that the enthalpy of the dehydration reaction, in vacuum, to be -131.3 kJ/mol (which includes the enthalpy of desorption of the water molecule from the surface).

6. The negative charge of the GO sheet in water might be highly dependent on water pH at the beginning of the simulation. The H^+ electrochemical potential in water is low when the simulation starts (there is no H^+ in solution) consequently there might be a high drive for the dissolution of the most movable species in GO following the prototypical reaction:

Which will continue until the chemical potential of the adsorbed H and the sum of electrochemical potentials of e^- and H^+ are equal. This leads to an electrification of the GO sheet that is highly dependent on the initial density of H^+ in solution. How do Authors comment about this dependency?

We agree with the reviewer that there will be an impact of the presence of protons (and more generally, of any solute) in the water. The deprotonation of GO surface groups, in particular alcoholate, is — as the reviewer has written — an acid-base equilibrium, which will be displaced depending on the concentration of H^+ in the liquid water (i.e., the pH). This equilibrium is typically well quantified in experimental work, through the measurement of pKa values of the graphene oxide (see for example Orth et al, *J. Colloid. Interf. Sci.*, **2016**). In *ab initio* simulations, the calculation of pKa values is, on the contrary, very challenging and an active topic of research in itself.

In this work, we aimed at studying the effect of pure water (with no solute and no self-ions) as a starting point: it is necessary to understand it well before introducing further species. We aim, in the future, at studying the effect of solutes in the water, in particular the possible presence of H^+ and HO^- ions (to understand possible pH dependence) as well as other ions (such as Na^+ and Cl^- , which are particularly relevant for applications in e.g. desalination). We mention this as an important perspective in the conclusion of our manuscript (page 11).

REVIEWERS' COMMENTS:

Reviewer #1 (Remarks to the Author):

I think that the authors have well addressed my comments and that the revisions are appropriate.
I recommend the publication of this manuscript in Nature Communications.

Reviewer #3 (Remarks to the Author):

The Authors replied to the questions in a satisfactory way within the limitations of their computationally expensive method.
I would recommend publication on Nature Communications

POWER PDF Trial
www.kotax.com

Point-by-point response to the reviewers

Reviewer #1

This work reports very interesting results concerning the structure of graphene oxide surfaces and their interactions with liquid water. The manuscript is very well written, and the results will be of interest for a large audience since such materials are investigated for many applications. I recommend publication in Nature Communications but I think the following points should mandatorily be addressed by the authors:

1 (minor): I think that the term “for the first time” should be avoided in the abstract. In addition I do not really see which gap is bridged in the same sentence.

We have removed “for the first time”, and expanded the sentence to make it clearer in the abstract.

2 (major): About the construction of the graphene oxide model: References 17 and 18 seem to be about graphite oxide. Don't the authors expect differences with graphene oxide?

Indeed, references 17 and 18 discuss a graphite oxide model, based on experimental data from graphite oxide. However, they focus almost exclusively on the graphene oxide layer itself, and have been widely used in the literature as a reference model for both graphite oxide and graphene oxide. There are definitely structural and chemical differences between graphene oxide and graphene, as well as there are between graphene oxide with different chemical compositions and syntheses. The chemical composition we have taken in the model we developed in this work are, on the other hand, rather typical of graphene oxide.

We have added a note, after refs. 17 and 18 in the introduction part, to indicate that the Lerf and Klinowski models were originally developed for graphite oxide, and which has since then been largely used in the literature to describe monolayers of graphene oxide.

- About the choice of having a functionalization rate of 25%: in the introduction the authors mention that it is of 5-10% when a thermal exfoliation is applied. Isn't that case the most relevant one?

Indeed, the 25% of O/C ratio lies in the lower range of observed functionalization of GO samples at room T — but in the higher range compared to ratios observed after annealing. It should be pointed out that converting our O/C ratio into the number of C atoms affected by the O functionalization yields higher percentage, i.e. $(12 + 2 \times 6) = 24 / 72 = 0.33$ for the random case and $24/36 = 0.66$ (locally) for the semi-ordered case. Therefore, taking a model with a O/C ratio close to the higher limit (~ 0.375 corresponding to 18 OH and 9 epoxide that affect 27 C atoms) would have restricted the number of possible independent replicas for the very dense semi-ordered case (coverage of 0.75%).

On the other hand, taking a lower O/C ratio (closer to 5 %) would have made the impact of GO on the water smaller, requiring more replicas in order to measure statistically significant variations: both models (semi-ordered and random) would be closer in behavior.

We have added a comment in the model section : “We note that the choice of O/C ratio results from a compromise: taking values closer to the higher experimental limit would have decreased the number of possible independent replicas (for the semi-ordered case because the number of C atoms bonded to an oxygen atoms acquire the majority status); while a lower ratio would require more simulation replicas in order to measure statistically significant changes in behavior.”

- Page 9, when comparing their results with ref. 45, the authors mention the potential presence of carboxylic functions. Why didn't they consider them in their model?

This was done for the sake of simplicity: carboxylic groups are known experimentally to be in smaller concentration than –OH and epoxides. Moreover, carboxylic functions are present only at the edges of the graphene oxide sheet, and we have chosen not to create edges in our periodic models — introducing explicit edges in the model would make its size much larger than it currently is, and difficult to study with sufficiently long MD trajectories. Or, done on a smaller model, it would create an unrealistic edge/surface ratio, misrepresenting the actual chemistry of the graphene oxide.

We have added a comment on the draft in the model section: “By construction with periodic systems, we discard the edge functionalization with carboxylic functions, which are experimentally known to be in smaller proportion, as compared to hydroxyls and epoxide ones.”

3 (major): About the validation of the graphene oxide model: unless I missed something, the only argument for preferring semi-ordered structures over the random ones is based on an energy criterion. However I see in Table 1 that the difference is not so large, and in fact random² has an energy lower than 3 of the five semi-ordered structures. I can imagine that there are infinities of structure that can be generated with the same level of defects (and this would increase exponentially with the size of the sample, which here remains quite small – even if it is large for DFT calculations!), how can the authors be sure that their data is enough to reach this conclusion? Wouldn't there be some experimental data (Raman, infra-red) that would allow them to have a more quantitative proof?

We agree with the referee that the data obtained within this work is not sufficient, in and of itself, to reach the conclusion that energetic stability is systematically favorable for semi-ordered structures. We note that this direct observation of the energetics on realistic models by first-principle methods is consistent with the results obtained theoretically and computationally by other methods, that we quote in our revised manuscript. On this, see also the discussion to reviewer #3, point 1).

This comment, as well as the remarks from reviewer #3 (point 2) have also prompted us to look into the impact of hydration on the energetic stability of our GO models: the extra analysis performed was added to the supplementary information (Table S2) and is discussed below.

4 (major) Page 8, last paragraph: The authors conclude that the different replicas behave very differently. I think that this conclusion should be mitigated in view of the very short simulation times. Are they sure that similar reactions would occur if they used different starting points for each of the simulations?

This is a very interesting remark, and it has prompted us to consider the impact of the initial configuration of the water molecules on the properties of the hydrated graphene oxide models. In order to probe this effect directly, we have performed additional *ab initio* MD simulations on one of our models (semi-ordered model 3). For this GO model, we have applied our methodology a second time, generating a different random initial packing of the same number of water molecules. We have added these results in Supplemental Information (Figure S8 and related discussion). The main conclusion is that, although the details of the dynamics are different (because the initial configurations are different), the overall behavior is rather close: opening of one epoxide and associated formation of one surface alcoholate on average.

Reviewer #2

Bocquet and co-workers presented a computational study, using *ab initio* calculations, to study the structure

and chemistry of graphene oxide (GO) in liquid water. This work studied GO in detail, and found that (i) the clustered (de-noted semi-ordered) GOs are more stable, relative to the random ones and (2) the similar/dissimilar structure behaviors including hydrogen bonding environments for studied GO structures in water, (iii) the fast diffusion of water on the GO surface, and (iv) the surface reactivity and its formed negative net charge. I think this work is well written and is evidently conducted with care. While there are still a few points that I think the authors should consider and address, my major concern is regarding the breakthrough achieved in this work to warrant a publication in *Nat. Commun.* Although this work helps bridge the gap in the understanding of GO materials, I do not see a breakthrough from this work to facilitate the development of GO in applications. I am therefore afraid that I do not find this work to be suitable to publish in *Nat. Commun.*, but in a good physical journal such as *JPC B* or *C*. My recommendation is to reject this submission and suggest the authors to submit to other journals as suggested above.

A few other comments:

1. It would be important to also study the effect of oxygen concentration (number of functional groups and their ratios).

The computational methodology we have developed in this work, and presented in the manuscript, can be applied to systems with slightly different characteristics: different O/C ratio, different ratio of epoxides/OH groups, presence of carboxylic groups (see discussion of this in response to reviewer #1's comment). However, these first-principles MD simulations are computationally costly, because they describe the system at the quantum chemical level (with DFT) and sample the dynamics of the systems studied. We therefore needed to make a choice in the systems under study, and have preferred in this initial study to focus our HPC time to obtain multiple replicas of each type of model (to make sure our results were robust), rather than test different chemical compositions.

We have added some discussion in the conclusion about the perspectives opened by this work, and in particular, about the interest in studying the influence of O/C ratio and in particular the low O/C ratio around 5%.

2. The authors commented on the stability based on the vacuum condition. Would the conclusion change if it is under solvated environment? Grossman and co-workers reported many years ago some evidences of "clustering of functional groups on GO/rGO materials (after annealing)". Maybe the results are in line with their observations?

We agree with the referee that the results obtained in this work on energetic stability, done in vacuum, do not preclude the possibility of an effect of the solvation environment. What we have seen in our simulations is that the presence of solvent, and the relaxation of the GO sheet that it induces, does not change the overall conclusions regarding stability. In this, our results agree with the findings of Grossman (DOI: 10.1016/j.carbon.2015.12.087).

We have added this reference, and related discussion, to the main text of the manuscript (page 4). We have also added a comparison of energies under solvation in the Supplementary Information (Table S2).

3. It remains unclear how the surface reactivity relates to the detailed structure properties of GO. More discussion would make this work stronger.

We thank the reviewer for this suggestions, which is also found in some comments of the two other reviewers. We have identified a specific structural motif with reactive properties, namely the presence of three consecutive -OH groups, pointing toward the same direction, and linked by hydrogen bonds. This motif gives rise to dehydration reaction of the middle -OH group, seen both in vacuum and facilitated in the hydrated GO.

4. What's the water diffusivity on the surface of random GO structures? Also, why the semi-ordered 4 has a noticeably smaller diffusivity than the semi-ordered 1 structure? Are water molecules mainly diffusing through the non-functionalized (sp^2) part?

We have tried to find a simple chemical explanation for the difference in water diffusion between the different semi-ordered models. We have looked at the analysis of the structures themselves and the presence of intramolecular H bonds (Table S1), as well as the number of non-functionalized graphene rings and diameters of the non-functionalized regions. We have not been able to establish any direct correlation between those parameters and the diffusion coefficients. Moreover, since diffusion is a collective phenomenon, we cannot separate the contributions from the water molecules near the non-functionalized sp^2 regions.

Moreover, given the relative short simulations times imposed by the high computational cost of ab initio molecular dynamics, we do not want to overinterpret the differences between the semi-ordered models, but focus on the average picture, which is the relatively fast dynamics of the water near the GO sheets, in comparison with the slowdown expected at this level of confinement.

Reviewer #3

The manuscript by F. Mouhat et al. reports the study of GO layers in water by means of ab initio molecular dynamics. The Authors find that GO sheets showing pristine graphene areas next to clusters of functional groups are the most stable configurations. Additionally, the GO layer is said to acquire a negative charge when immersed in water as a consequence of the release of protons. The manuscript provides interesting insights into GO atomistic structure and reactivity. However, some of the conclusions (the clustering effect) were already proven to be independent on the surrounding media in literature while others (some reaction paths) might be also drawn for a sheet in absence of water. For this reason, the difference between GO in vacuum and in water should be highlighted with more explicit comparisons. The manuscript provides fertile ground for new very appealing analyses which might be implemented to enhance the impact of the paper. At the current state I would not recommend the manuscript for Nature Communications, but I would suggest major revisions.

Here are some comments on how to improve the quality of the manuscript:

1. One of the main points of the manuscript, the clusterization of the functional groups, was already proven both experimentally and theoretically [see the experimental study from Kumar et al. Nat. Chem. 6, 151 (2014) and the theoretical works: Carbon 100, 90 (2016), Sci. Rep. 3, 2484 (2013)] The main reason behind this behaviour is attributed, as indicated also by the Authors, to the strain induced by functional groups that is released if the functional groups move together in a suitable conformation [see, for instance, PRL 96, 176101 (2006)]. As such, I would rather focus the study on the effect that the presence of water has in the clustering process and I would perform more analyses about this.

We agree with the reviewer that the existence of clustering of oxygen groups is not new, but would note that from the computational point of view, they have been previously studied in most cases with simpler cluster models at the DFT level (Zhou & Bongiorno, *Sci. Rep.* 2013), or with less accurate reactive force fields (Kumar et al, *Carbon* 2016). Moreover, although we are satisfied to find this behavior reproduced, we mostly use it as a starting point for further analyses of the properties of the hydrated GO (which were further extended as part of the review process).

We have added a sentence to the discussion on page 4 about the clustering of oxygen group, to highlight the previous works on this topic, as suggested by the reviewer.

2. In relation to the previous point, the Authors suggest a very interesting hypothesis on another possible drive for the clusterization: the maximization of the number of hydrogen bonds. This hypothesis deserves a more detailed analysis as it can be the key to understand the effect of water on GO. I would expect that the presence of water hinders the tendency for the functional groups to cluster as, when the sheet is submerged in water, hydrogen bonds form between the sheet and the H₂O molecules, rather than only among the GO functional groups.

A more detailed investigation on the water effect would quantify the consequences of intermolecular H bonding as opposite to intramolecular as, here, might lie the core for a different clustering process.

We thank the reviewer for this suggestion, which is very interesting. Indeed, we had not initially looked at the impact of hydration on the energetics of the GO models. We have performed further calculations, and have inserted a new Table S2 to the Supplementary Information. This analysis shows that the relative energetic stability of random vs. semi-ordered models is not changed upon hydration. We find that the energy difference (on average) between semi-ordered and random models is changed from 2.45 eV (without water) to 3.27 eV (hydrated).

We have added a few sentences about this analysis in the “Structural properties” section, on page 5.

3. Authors should provide a direct comparison with their trajectories in vacuum as many of the chemical reactions occurring do not involve water molecules directly. For instance, the reactions described in Figure 3a and 4 might as well take place in vacuum.

This is a very valid point, and in order to probe the behavior of our GO models in vacuum, we have run them further for 60 ps of production dynamics. These longer simulations have revealed a very interesting behavior: the reactive motif identified in semi-ordered model 4 can also lead to dehydration (loss of a water molecule) in the GO in vacuum, but that event only occurs after 40 ps of dynamics, while the reaction is fast in the hydrated model. We discuss this comparison of hydrated GO and empty GO in the revised manuscript, on page 9, and in the supplementary information (Figure S3 and related discussion, page S7).

4. I have some concerns about the methodology for the random generation of GO sheets. It was proven that epoxides and hydroxyls cluster in specific arrangements that lower the sheet energy [see Sci. Rep. 3, 2484 (2013) and J. Phys. Chem. Lett. 10, 7492 (2019)]. Could the random generation lead to highly unstable sheets that might enhance the likeliness for functional groups to desorb? For example, the reaction in Figure 4 that leads to a reduction of the sheet might be caused by the excessive strain on the GO sheet induced by the nearby functional groups. What's Authors' opinion in this regard?

The randomness in the generation of the models is, we think, a key component in order to make sure that the models are as realistic as possible, i.e. that we do not introduce specific features artificially. We agree with the reviewer that, because of this random element in the generation of models, it could lead in some cases to highly unstable GO sheets. We have, however, taken steps to avoid this: first, only models that are stable upon initial energy minimization have been considered (i.e., the models are not spontaneously reactive). Secondly, we have performed MD simulations only on some, and not all, of the models generated: we selected these models by energetic criteria, therefore eliminating potentially highly unstable model. Finally, we have run simulations on several models of each type (replicas) in order to study the diversity in behavior, and to obtain physically meaningful averages that do not depend only on a single simulation.

5. The different mechanisms shown in the paper are very interesting and deserve a deeper study, first of all, to identify whether there is an effect of the surrounding functional groups and secondly to understand what are the conditions that lead to such transitions. Some non-dynamical studies about energy and barriers could reveal more information.

Using both the simulations of the hydrated models, and the new simulations of the models in vacuum, we have been able to identify a specific reactive motif in the GO structure, that leads to the dehydration reaction. We have marked it in Figure 1, and discuss it in the main text on page 9, and in the supplementary information (page S7).

As suggested by the reviewer, we have used the MD simulations to estimate the energetics of the dehydration reaction. The results were added to the supplementary information (page S9 and Figure S4). We find that the enthalpy of the dehydration reaction, in vacuum, to be -131.3 kJ/mol (which includes the enthalpy of desorption of the water molecule from the surface).

6. The negative charge of the GO sheet in water might be highly dependent on water pH at the beginning of the simulation. The H^+ electrochemical potential in water is low when the simulation starts (there is no H^+ in solution) consequently there might be a high drive for the dissolution of the most movable species in GO following the prototypical reaction:

Which will continue until the chemical potential of the adsorbed H and the sum of electrochemical potentials of e^- and H^+ are equal. This leads to an electrification of the GO sheet that is highly dependent on the initial density of H^+ in solution. How do Authors comment about this dependency?

We agree with the reviewer that there will be an impact of the presence of protons (and more generally, of any solute) in the water. The deprotonation of GO surface groups, in particular alcoholate, is — as the reviewer has written — an acid-base equilibrium, which will be displaced depending on the concentration of H^+ in the liquid water (i.e., the pH). This equilibrium is typically well quantified in experimental work, through the measurement of pKa values of the graphene oxide (see for example Orth et al, *J. Colloid. Interf. Sci.*, **2016**). In *ab initio* simulations, the calculation of pKa values is, on the contrary, very challenging and an active topic of research in itself.

In this work, we aimed at studying the effect of pure water (with no solute and no self-ions) as a starting point: it is necessary to understand it well before introducing further species. We aim, in the future, at studying the effect of solutes in the water, in particular the possible presence of H^+ and HO^- ions (to understand possible pH dependence) as well as other ions (such as Na^+ and Cl^- , which are particularly relevant for applications in e.g. desalination). We mention this as an important perspective in the conclusion of our manuscript (page 11).